# Corticosteroid Usage in Modeling Gulf War Illness in Pre-Clinical Models: A Systematic Review

**DOI:** 10.3390/ijms262110269

**Published:** 2025-10-22

**Authors:** Lily Tehrani, Chetana Movva, Joshua Frank, Stephanie Nagy, Riya Davar, Bhumika Balani, Nancy G. Klimas, Lubov Nathanson

**Affiliations:** 1Institute for Neuro-Immune Medicine, Dr. Kiran C. Patel College of Osteopathic Medicine, Nova Southeastern University, Fort Lauderdale, FL 33328, USA; lt1125@mynsu.nova.edu (L.T.); jf2048@mynsu.nova.edu (J.F.); sn903@mynsu.nova.edu (S.N.); nklimas@nova.edu (N.G.K.); 2Halmos College of Arts and Sciences, Nova Southeastern University, Fort Lauderdale, FL 33328, USA; cm3674@mynsu.nova.edu (C.M.);

**Keywords:** Gulf War Illness (GWI), corticosteroids (CORT), neuroinflammation, immune suppression, pro-inflammatory cytokines, oxidative stress, epigenetic changes, HPA axis dysfunction, cognitive impairment neurotoxicity

## Abstract

Gulf War Illness (GWI) is a neuroinflammation- and immune-dysfunction-related chronic disease. Corticosteroids, a class of steroid hormones with potent anti-inflammatory and immunosuppressive properties, have been studied for their role in GWI pathophysiology. Eight corticosteroid effect studies were evaluated in this systematic review. Preclinical models showed exacerbation of neuroinflammation, oxidative stress, and epigenetic changes with exposure to CORT in addition to Gulf War neurotoxicants, which induced pro-inflammatory cytokine expression (Tumor Necrosis Factor Alpha (TNF-α), Interleukin-6 (IL-6), C-C motif chemokine ligand 2 (CCL2)). Such findings suggest that corticosteroids can exacerbate symptoms of GWI and need further clinical research to clarify their role in neuroinflammatory processes.

## 1. Introduction

Gulf War Illness (GWI) is a chronic, multi-symptom disorder affecting approximately 30% of the 750,000 U.S. veterans who served in the 1990–1991 Persian Gulf War [1]. Clinically, GWI presents a wide array of debilitating and chronic symptoms, such as chronic fatigue, sleep disturbances, neurocognitive symptoms, lung diseases, gastrointestinal disturbances, skin lesions, and joint pain and stiffness [2]. Recent research has made strides in understanding the pathophysiological mechanisms of GWI. Promising experimental therapies, such as neuroprotectants and neurosteroids, have been identified for their neuroprotective and disease-modifying properties [3]. Despite the significant burden of disease, there are currently no FDA-approved treatments for GWI, and accelerated biomedical and clinical studies are urgently needed to begin effective treatment [3,4].

Exposure to a wide range of toxic environmental substances during deployment has been strongly associated with the pathophysiology of GWI [5]. These exposures include nerve agents like sarin gas, pyridostigmine bromide, organophosphate pesticides, and chlorinated hydrocarbons [2,6,7,8,9,10,11]. Studies suggest that these neurotoxic exposures may trigger a cascade of chronic neuroinflammatory and immunological disturbances, contributing to the prolonged symptomatology observed in affected veterans [8]. A distinct proinflammatory biomarker profile has been associated with GWI in various studies [12]. Increasing evidence also supports the role of neuroinflammation, oxidative stress, and dysregulated immune responses in GWI, with findings demonstrating persistent glial activation and elevated proinflammatory cytokine levels in both preclinical and human studies [13,14].

In recent years, this inflammatory response in GWI veterans has been a widely considered area of exploration. Within the body, inflammatory responses can be changed in part by corticosteroids, which are a broad class of steroid hormones that include both glucocorticoids and mineralocorticoids [15,16,17].

Corticosteroids influence inflammatory immune responses through vasoconstriction, decreased chemotaxis, and macrophage interference [18]. Glucocorticoids are a cornerstone of anti-inflammatory and immunosuppressive treatment, universally effective in the therapy of acute and chronic inflammation. There are many documented therapeutic applications, and they cover a spectrum from diseases like rheumatoid arthritis, inflammatory bowel disease, and psoriasis to use in immunosuppressive regimens following organ transplantation [19,20,21,22]. The therapeutic effects of glucocorticoids are mediated through multiple mechanisms, such as inhibition of initial inflammatory responses like vasodilation and suppression of genes encoding pro-inflammatory cytokines and chemokines [23].

Corticosteroids also have the unique ability to induce inflammatory responses, as observed in preclinical models [24]. To better understand the pathophysiology of GWI, preclinical models have utilized corticosteroids to induce chronic inflammation, replicating key disease characteristics seen in GWI veterans. Many of these models rely on corticosterone (CORT), the rodent analog of cortisol, to induce a state of chronic stress and exacerbate neuroinflammation [25]. Findings from preclinical models suggest that stress and immune dysregulation may play a central role in exacerbating GWI-related symptoms.

This systematic review aims to consolidate current findings on the role of corticosteroids in GWI, critically evaluating preclinical studies to understand their impact on disease pathophysiology. By analyzing the existing literature, we seek to determine the role of corticosteroids in GWI research and to identify gaps in knowledge that warrant further investigation. This question is clinically relevant because GWI affects nearly one-third of Gulf War veterans, yet no FDA-approved treatments currently exist. Neurotoxicant exposures implicated in GWI drive persistent neuroinflammation, a process that directly intersects with corticosteroid signaling. While corticosteroids are clinically recognized for their anti-inflammatory and immunosuppressive effects, preclinical GWI models reveal paradoxical findings in which corticosteroid exposure can exacerbate neuroinflammation, oxidative stress, and cytokine expression. Exploring this bidirectional role is essential not only for guiding rational therapeutic development but also for identifying targetable pathways and addressing critical knowledge gaps that currently impede progress in treating this condition. Understanding the complex relationship between corticosteroids, neuroinflammation, and immune dysregulation in GWI will serve as an essential guide in future research to develop targeted therapies that address the pathophysiology and severity of this condition.

## 2. Methods

A systematic literature review was performed using Ovid MEDLINE, Excerpta Medica database (EMBASE), and Web of Science. The search was conducted using the terms “Corticosteroids AND (Gulf War Illness OR GWI)” and “Corticosterone AND (Gulf War Illness OR GWI).” All articles that were published after 1990 and written in the English language were included. The relevance of the articles was evaluated in a hierarchical approach in which first the title and abstract were assessed, and then the full texts were analyzed to ensure they fit the scope of the article. For articles that were not freely accessible, the Nova Southeastern University library database was used to gain access.

The studies were considered eligible if they provided relevant data on the use of corticosteroids in the development and/or treatment of GWI. Eligible study designs included randomized clinical trials, experimental animal models, and case–control studies. Inclusion criteria involved free full-text accessibility, articles published after 1990, articles in the English language, and studies relevant to the relationship between corticosteroids and GWI. Due to the novelty of research conducted within GWI, the years included were extended to ensure all possible studies conducted in the field were included in this review. Exclusion criteria comprised reviews (scoping, systematic or literature reviews), inaccessibility of the full-text article, articles published prior to 1990, or if the research was unrelated to GWI or the effects of corticosteroids. Each article was screened by two researchers in a blind review process, with a third researcher mitigating any disagreements. A flow diagram of the selection criteria (Figure 1) was developed using the requirements outlined by Preferred Reporting Items for Systematic Reviews and Meta-Analyses (PRISMA) [26].

## 3. Results

In total, 212 articles were extracted from the databases Ovid MEDLINE, EMBASE, and Web of Science based on keyword searches. After the first-tier screening, 123 duplicate articles were removed, and an additional 70 articles were also excluded based on title and abstract, lack of full-text availability, study type, and publication year before 1990. Review articles were all excluded. Additionally, if the study was irrelevant to corticosteroids and any possible relationship with GWI, it was excluded. Eighteen articles that remained after the first-tier screening underwent full-text screening. During tier II review, eight studies were excluded, five of which were found to not provide relevant data on corticosteroids in the context of GWI treatment or management. The other four articles were unrelated to GWI and its relationship to corticosteroids. After the full-text screening (tier II review), only eight articles remained to be analyzed for eligibility and inclusion into the study. Table 1 depicts the studies included in the analysis between GWI and corticosteroids.

To evaluate if CORT exposure primes the central nervous system and increases neuroinflammation, an experimental animal model was used in which male mice were injected subcutaneously with pyridostigmine bromide (PB) (2 mg/kg/day) and *N,N*-diethyl-meta-toluamide (DEET) (30 mg/kg/day) for 14 days [27]. From days eight to 15, the experimental group of rats received CORT in their drinking water (200 mg/L in 1.2% EtoH) while control groups received saline injections and a parallel 1.2% EtoH/water vehicle. On day 15, the experimental group received a single injection of diisopropyl fluorophosphate (DFP) (4 mg/kg) while the control group received saline (0.9%). Another group received anti-inflammatory pre-treatment with minocycline (MINO), an anti-inflammatory antibiotic. In this group, mice were given 100 mg/kg of MINO subcutaneously for 15 days. Researchers found that DFP alone led to widespread neuroinflammation across multiple brain regions. CORT pre-treatment exacerbated DFP neuroinflammation in all brain regions, while PB/DEET alone reduced DFP-induced inflammation. However, this protective effect was lost when combined with CORT, which continued to worsen the inflammation induced by DFP. Researchers also found that pre-treatment with MINO reduced neuroinflammation induced by CORT + DFP exposure, as seen by the reduction in proinflammatory markers. Together, these findings indicate that although GWI is associated with widespread inflammation, MINO might serve as a potential therapeutic intervention.

A subsequent study examined the effects of various neurotoxicants in an animal model by exposing male mice to CORT in their drinking water (400 mg/L in 1.2% EtOH) for four days followed by either a single 8 mg/kg dose of intraperitoneal injection (IPI) of chlorpyrifos oxon (CPO), 4 mg/kg IPI of DFP, 3 mg/kg IPI of PB, or 0.5 mg/kg IPI of physostigmine (PHY) [28]. Exposure to DFP, an irreversible acetylcholinesterase (AchE) inhibitor, led to an increase in the expression of pro-inflammatory cytokines CCL2 and TNF-alpha in the cortex and hippocampus. Pretreatment with CORT exacerbates these levels, with increases in all six inflammatory cytokines that contribute to neuroinflammation. Similarly, solo CPO administration increased the expression of proinflammatory cytokines while pre-treatment with CORT also markedly increased CPO-induced neuroinflammation. Reversible AchE inhibitors, such as PB, did not produce neuroinflammation and acted rather as an anti-inflammatory. Additionally, the proinflammatory cytokine expression in CORT + DFP and CORT + CPO exposed mice activated the JAK/STAT3 pathway, where STAT3 is a neuroinflammatory signal transducer. These findings confirm the fact that DFP and CPO increase neuroinflammation following CORT exposure, since PHY and PB alone (with or without CORT) did not trigger neuroinflammation or STAT3 activation. These findings are consistent with brain studies showing that gp130-JAK2-STAT3 is activated early in injury, drives reactive gliosis and neuroinflammatory gene programs, and that inhibiting this pathway in glia reduces astrogliosis and downstream cytokine signaling [34,35,36,37,38]. Researchers concluded that when rodents were treated with AchE inhibitors like DFP, CPO, and PHY, AchE activity decreased. However, when rodents were pretreated with CORT, this reduction in AchE activity for the irreversible inhibitors (DFP and CPO) was prevented. This pattern supports that CORT exacerbates neuroinflammation through mechanisms independent of cholinergic inhibition, likely involving alternative molecular pathways such as JAK/STAT3 signaling activated by organophosphorylation.

To further characterize the molecular effects of CORT and DFP exposure, an experimental animal model was used to examine how prior stress exposure influences neuroinflammatory and epigenetic responses to a sarin surrogate [29]. Seventy-nine adult male mice were divided into four experimental groups: Group 1 was given saline in the drinking water for four days, followed by an IPI of saline (0.9%) on day five. Group 2 was given CORT in drinking water (200 mg/L in 0.6% ethanol, EtoH) for four days, followed by an IP of saline (0.9%) on day five. Group 3 received DFP in the drinking water for four days, followed by a DFP IPI on day five. Lastly, group 4 was given CORT for four days in the drinking water, followed by an IPI of DFP (4 mg/kg) on day five. All mice groups were sacrificed six hours later, and the transcriptome was analyzed using RNA sequencing (RNA-Seq) while the epigenome was analyzed using reduced representation bisulfite sequencing (RRBS) and H3K27ac ChIP-seq.

RNA-seq analysis of the frontal cortex of CORT + DFP exposed mice revealed 206 unique differentially expressed genes (DEG) that were linked to oxidative stress, steroid biosynthesis, and the immune response. A similar analysis of the hippocampus of CORT + DFP exposed mice revealed 667 unique DEGs that were linked to immune signaling, neuronal development, and differentiation. These results indicate that there are overlapping pathways and genes between the hippocampus and frontal cortex, highlighting the shared effects that result from combined exposure to CORT + DFP. DNA methylation analysis of the frontal cortex revealed 297 differentially methylated cytosines across 60 regions, but no significant biological patterns were linked to these changes. Analysis of the hippocampus revealed more extensive modifications as 926 differentially methylated cytosines were found across 192 regions. Enrichment analyses revealed that these changes were linked to norepinephrine metabolism and cilium structure. Upon estimating cell proportions in the cortex, the investigators found that CORT exposure, with or without DFP exposure, increased the number of neurons and decreased the number of myelinating oligodendrocytes. Chromatin accessibility was assessed by measuring H3K27ac, a marker of active transcription in the genome. 3294 genes had changes in chromatin accessibility in the frontal cortex, whereas 1518 had similar changes in the hippocampus. Enrichment analyses revealed a bias toward neuronal-related pathways including neuronal morphology and synaptic signaling pathways. Collectively, these findings suggest that co-exposure to CORT and DFP alters transcriptional pathways, histone modifications, and DNA methylation, leading to gene expression changes that closely resemble those observed in GWI veterans.

To evaluate whether these molecular changes corresponded with structural alterations, a rat model was used to assess how prior CORT exposure influences DFP-induced neuroinflammation and brain microstructure [30]. In this study, male rats received CORT in their drinking water (200 mg/L in 0.6% EtOH) for four days, followed by an injection of DFP (1.5 mg/kg). Real-time PCR was utilized to measure pro-inflammatory cytokine expression while diffusion magnetic resonance imaging (dMRI) was used to assess brain microstructure. Rodents that were exposed to DFP alone revealed an increased expression of proinflammatory cytokine genes, such as TNF-alpha and IL-6; however, the ones that had prior exposure to CORT had a significantly exacerbated cytokine response. Micro-scale diffusivity mapping revealed higher values in the CORT, DFP, and CORT + DFP groups compared to the controls. The diffusion encoding length at 10 lumens (lm) was able to detect statistically significant changes between the groups. CORT + DFP treated rodents had larger clusters identified on imaging in the hippocampus and outer cortex. The most pronounced diffusivity values were observed in the thalamus, amygdala, piriform cortex, and ventral tegmental area of rats that were exposed to DFP (*p*< 0.001) followed by the ones exposed to CORT (*p* < 0.01). These findings highlight that prior CORT exposure enhances both inflammatory and structural brain responses to DFP, suggesting that similar neuroinflammatory and microstructural alterations may underlie GWI pathology.

To compare central and peripheral inflammatory responses, another study evaluated the differential effects of organophosphates and CORT on the brain versus peripheral tissues by measuring cytokine levels in the liver and serum [39]. Adult male mice were given a single subcutaneous injection of PB (2 mg/kg/day) and DEET (20 mg/kg/day), each day for 14 days. On day eight, the mice were given CORT (200 mg/L in 0.6% EtoH) in their drinking water for seven days. An IPI of DFP (4 mg/kg) was given to the mice on day 15, and they were sacrificed by decapitation post-injection at 2, 6, 12, and 72 h. Exposure to PB or co-exposure with DEET showed either little to no proinflammatory effects in the brain and reduced levels of liver mRNA cytokines compared to groups that were not given PB/DEET pretreatment. DFP alone, however, increased the mRNA expression of the proinflammatory cytokines IL-6, IL-1B, leukemia inhibitory factor (LIF), and oncostatin M (OSM) in the blood and liver at 6 h post-exposure, and increased levels of IL-10, an anti-inflammatory cytokine. Surprisingly, CORT exposure reduced the DFP-induced pro-inflammatory cytokine expression, except for LIF. However, CORT exposure exacerbated DFP-induced neuroinflammation in the brain. The findings indicate that neuroinflammatory pathways are distinct from peripheral cytokine responses, likely reflecting brain-specific inflammatory priming by CORT.

Expanding the analysis to gene networks and signaling pathways, another study examined how combined exposure to CORT and DFP affects gene expression and signaling pathways in an experimental rat model [31]. Male and female rats were divided into three groups: the control group, the DFP group, and the CORT + DFP group. The control group received tap water for fluid and a saline injection, while the DFP group received plain tap water for fluid and was then followed by IPI with DFP (4 mg/kg). The CORT + DFP group received tap water that contained CORT (20 mg/L in 0.6% EtoH) and then an IPI with DFP (4 mg/kg) on day eight. Combined CORT + DFP exposure significantly activated the cytokine–cytokine receptor pathway and the TNF signaling pathway in comparison to the control and DFP group. Protein–protein interaction analysis identified 6 subnetworks for the DEGs in the CORT + DFP group, including key regulators Cxcl1, IL-6, Ccnb1, TNF, Agt, and Itgam. These subnetworks function in multiple pathways, including the cytokine–cytokine receptor interaction, p53 signaling, neuroactive ligand–receptor interaction, and the complement and coagulation cascade. The quantitative trait locus (QTL) of DEGs revealed 21 DEGs in the CORT + DFP group. These links were not observed in the control or DFP group, indicating specific genome regions involved in the response to CORT + DFP. The gene A Disintegrin and Metalloproteinase with Thrombospondin motifs 9 (*Adamts9*) was identified as a potential molecular link between GWI and Chronic Fatigue Syndrome (CFS). *Adamts9* expression was significantly altered following combined exposure to CORT and DFP, suggesting its involvement in GWI pathogenesis [31]. This connection is supported by prior genetic studies associating *Adamts9* with inflammation and cognitive aging, mechanisms that parallel the dysfunction observed in GWI [40,41,42,43]. Furthermore, the biological plausibility of this gene’s role is supported by the function of its protein product, ADAMTS9, a secreted metalloprotease essential for maintaining the extracellular matrix and inhibiting angiogenesis [43].

To explore potential therapeutic modulation of neuroinflammation, a long-term mouse model was used to test whether propranolol, a β-adrenergic blocker, could reduce inflammation induced by repeated CORT and DFP exposure [32]. Adult mice were exposed to CORT (200 mg/L in 0.6% EtOH) in their drinking water for seven days, followed by either an IPI of DFP (4 mg/kg) or saline on day seven. CORT exposure was then repeated every other week for a total of five weeks to promote a chronically primed neuroinflammatory phenotype. On day 35, mice received a subcutaneous injection of lipopolysaccharide (LPS) (0.5 mg/kg) or saline. A subset of animals received propranolol (20 mg/kg in saline) either four days before the LPS challenge (day 24) or during CORT exposure (day 31). Six hours after LPS administration, there was a statistically significant increase in the expression levels of proinflammatory cytokine mRNA in both the cortex and hippocampus (*p* < 0.05). The CORT + DFP + LPS group had significantly increased cytokine mRNA expression in comparison to the CORT + LPS group. This suggests that exposure to GWI neurotoxicants, such as DFP, can significantly enhance the cytokine response upon immune challenge with LPS when introduced after priming the models with corticosterone. Administration of propranolol four days before the LPS challenge significantly reduced the neuroinflammatory response in both the cortex and hippocampus. However, when propranolol was given outside of CORT exposure, it provided no protective effects and significantly increased the expression of inflammatory cytokines in the CORT + LPS group. These findings suggest that propranolol can reduce CORT-enhanced neuroinflammation when administered during stress exposure, highlighting its potential as a therapeutic strategy for GWI-related neuroinflammation.

To further examine these long-term effects, a chronic exposure rat model was used to investigate how combined CORT and DFP exposure influenced neuroinflammatory responses to immune challenge [33]. Sixty-eight male rats were exposed to CORT (200 mg/L in 0.6% EtOH) in their drinking water for seven days, followed by a single IPI of DFP (1.5 mg/kg) on day eight. Animals were then re-exposed to CORT every other week for four weeks followed by a subcutaneous LPS challenge (0.25 mg/kg) on day 36. Six hours after LPS exposure, brain and liver tissues were collected and frozen for mRNA analysis, and after 24 h, whole brains were collected for MRI and immunohistochemistry. Following LPS exposure, the CORT + DFP + LPS group had higher levels of several pro-inflammatory cytokines, including TNF, IL-6, CCL2, and OSM, in both the hippocampus and cortex compared to the saline, CORT, and CORT + DFP group. Additionally, this group showed an elevated ratio of activated glial cells in the dorsal hippocampus and a higher generalized fractional anisotropy (GFA). These findings indicate that prior CORT + DFP exposure can exacerbate the cytokine response to immune challenge, leading to increased inflammation. Collectively, these results highlight the role of neuroinflammation and glial activation in the pathophysiology of GWI and suggest potential targets for future therapeutic interventions.

## 4. Discussion

### 4.1. Neuroinflammation Is a Central Mechanism

Neuroinflammation represents one of the important mechanisms in GWI pathophysiology. Indeed, combined CORT and DFP treatments in pre-clinical animal models increased expression of pro-inflammatory cytokines like TNF-alpha, IL-6, and CCL2, while also increasing the activation of glial cells, as presented by several studies including Ashbrook et al. [29], Cheng et al. [33], and Koo et al. [30]. For instance, Koo et al. [30] demonstrated that CORT exposure enhanced neuroinflammatory cytokine responses in rodents, a finding further supported by Cheng et al. [33], who reported that exposure to CORT caused enhanced inflammatory responses following subsequent immune challenges such as LPS. Various cytokines including TNF-alpha and IL-6 have been found to be involved in the immune profiles in studies involving human GWI subjects [44]. These findings agree with pre-clinical observations of chronic neuroinflammatory signatures in rodents.

### 4.2. Epigenetic and Transcriptional Changes

Studies that performed epigenetic and transcriptomic analyses emphasized the changes in gene expression that reflect neurological dysfunction associated with GWI. For instance, Ashbrook et al. [29] showed widespread chromatin remodeling and DNA methylation changes, especially within the hippocampus and frontal cortex. These regions are highly involved in cognition and emotional regulation capacities often disrupted in GWI. Enrichment analyses linked these modifications to oxidative stress, immune signaling, and neuronal differentiation pathways. Xu et al. [31] highlighted the cytokine–cytokine receptor and TNF signaling pathways as key mediators in the CORT + DFP group. Their findings note that Cxcl1, IL-6, and TNF are among notable DEGs. These genes, coupled with DEGs mentioned in studies outside of this review, including *TTll7*, *Slc44a4*, and *Rusc2* [45], suggest that epigenetic targets may be possible avenues for therapeutic intervention. Additionally, *Adamts9*, which is epigenetically regulated and differentially expressed in GWI models, is known to be involved in the remodeling of the extracellular matrix and the regulation of inflammatory reactions. Due to its role in tissue integrity and inflammation, it is also a potential biomarker and therapeutic target in GWI. Identification of *Adamts9* further points to a link with chronic fatigue syndrome and shared pathophysiology between GWI and other chronic inflammatory disorders.

### 4.3. Structural and Functional Brain Changes

Preclinical models using dMRI and FA analyses by Koo et al. [30] and Cheng et al. [33] suggest structural changes in key brain regions of the hippocampus and thalamus following exposure to CORT and DFP. These neuroimaging findings also reinforced neuroinflammatory changes and would point toward these structural changes and underlying cognitive and mood disturbances associated with GWI. These changes in structure could represent a non-invasive biomarker of disease progression and therapeutic efficacy in preclinical models.

### 4.4. Therapeutic Insights

These studies suggest several promising therapeutic avenues. Anti-inflammatory agents such as MINO [27] and propranolol [32] showed significant efficacy in reducing neuroinflammatory markers when administered during CORT or DFP exposure in preclinical trials. This suggests that targeting inflammation at specific windows of exposure may mitigate GWI symptoms. In addition, the JAK/STAT signaling pathway is a critical regulator of cytokine-induced immune responses. In GWI, abnormal activation of this pathway may be able to amplify neuroimmune signaling, perpetuating glial activation and inflammatory cytokine production that enhance neural damage and symptom chronicity. Dysregulation of JAK/STAT abnormally may thus be a central molecular mechanism for bridging peripheral immune signals to central nervous system dysfunction and hence potentially serve as a therapeutic target for modulation in GWI [28]. Additionally, the identification of key transcriptional regulators (IL-6, TNF, and Adamts9 [31]) provides additional therapeutic targets that may alleviate neuroinflammation and restore neural homeostasis.

### 4.5. Glucocorticoids in Human Studies

In human studies, endogenous glucocorticoids, such as corticosterone, are potent anti-inflammatory agents whose therapeutic power is well-documented in clinical medicine. At a molecular level, their primary action involves the repression of numerous genes that encode pro-inflammatory molecules like cytokines and chemokines. This effect is further enhanced by the local amplification of glucocorticoid action within inflamed tissues—a process mediated by the enzyme 11ß-hydroxysteroid dehydrogenase type 1 (11ß-HSD1), which increases the intracellular concentration of active glucocorticoids to promote the resolution of inflammation directly at the site where it occurs [23]. While this anti-inflammatory capacity is powerful in acute situations, it is important to note that prolonged exposure to elevated glucocorticoid levels, as seen in chronic stress, can paradoxically lead to pro-inflammatory effects [44].

The clinical efficacy of this anti-inflammatory mechanism is clear. In sepsis patients, for example, hydrocortisone (cortisol) effectively reduces pro-inflammatory cytokines while boosting the anti-inflammatory cytokine IL-10 [46]. Similarly, the body’s own endogenous cortisol has been shown to temper inflammatory responses, leading to a dose-dependent reduction of IL-6 after the acute inflammation of cardiac surgery [47]. This principle also applies to chronic inflammatory conditions, where even low-dose prednisone can significantly alleviate symptoms in patients with rheumatoid arthritis [48].

While these examples of glucocorticoid efficacy in conditions like sepsis, post-cardiac surgery inflammation, and rheumatoid arthritis are distinct from the specific pathologies of GWI, they collectively underscore the pervasive and well-established clinical use of glucocorticoids as potent anti-inflammatory treatments in humans. This widespread therapeutic application stands in stark contrast to their paradoxical role as exacerbators of inflammation in certain preclinical GWI models, particularly those employing stress-induced corticosterone.

While glucocorticoids are undeniably potent anti-inflammatory medications, their clinical application in humans requires meticulous consideration of dose, timing, and individual patient context. This precision is essential to maximize therapeutic benefit while minimizing the risk of adverse effects, which can be significant with long-term or high-dose use. However, further research is needed to better understand the appropriate application and potential mechanisms of glucocorticoids in clinical models and eventual therapies for GWI.

### 4.6. Limitations

Within this review, eight studies were analyzed to better understand the relationship between GWI and CORT. A noticeable observation from our analysis is the prevalent reliance on preclinical animal models, where corticosteroids were primarily utilized to model stress. This finding underscores a critical need for further research exploring the relationship between GWI and the therapeutic, clinical use of glucocorticoids in human populations.

Furthermore, while we adhered to a structured systematic review process to elucidate the complex relationships between GWI and glucocorticoids, we acknowledge that the included studies shared an institutional affiliation. While this pattern was noted, based on the rigorous methodology and reported findings of the research examined, we believe the presented results remain robust and accurately reflect the current body of evidence.

### 4.7. Model Validity and Limitations of Pre-Clinical Studies

It is important to note the eight pre-clinical studies reviewed in this study were designed to mimic certain features of GWI through combined exposure to stress hormones and neurotoxicants. These models successfully reproduce some of the inflammatory and immune changes seen in veterans, such as glial activation and elevated cytokines, which contribute to their face validity. However, they only approximate parts of the condition and cannot fully represent the range of symptoms and exposures that occur in humans.

Furthermore, the construct validity of these models is limited because stress is induced artificially through corticosterone exposure, the toxicant doses and timing cannot completely reflect what veterans experienced during deployment. Still, these methods provide a controlled way to study how stress and chemical exposure interact at the molecular and cellular levels. Similarly, the predictive validity of the models remains uncertain, since the medications that reduced inflammation in rodents, such as propranolol and MINO, have yet to show consistent benefit in human studies. The main strength of these models is in comparing how prior stress may alter vulnerability to toxicant exposure down the line, helping to understand why individuals from high-stress environments could have more severe and lasting effects than those from lower-stress conditions.

Collectively, these studies provide useful insight into how stress and toxicant exposure interact with GWI, but they should be viewed as supportive rather than conclusive. Continued research using different animal models and human studies is needed to confirm which findings truly apply to veterans with GWI.

## 5. Conclusions

The current body of literature presented here highlights the complex mechanisms behind the pathophysiology of GWI induced by neurotoxicants such as DFP and CORT in preclinical models of GWI. These findings deepen our understanding of the complex interplay between the associated changes, including neuroinflammation, transcriptional alterations, and brain structure changes, offering insight into the pathophysiology and future treatment targets in GWI.

These articles also suggest that neuroinflammation, transcriptional dysregulation, and structural changes in the brain could be linked to the pathophysiology of GWI and could be due to neurotoxicant exposures, such as DFP and CORT. Potential therapies to be developed could target neuroinflammatory pathways like JAK/STAT3 inhibition, and anti-inflammatory agents such as MINO appear promising but warrant further validation. These studies represent the groundwork from which treatments might be developed for the benefit of improving the quality of life for GWI veterans.

Together, these eight studies support a link between stress hormones and toxicant-related inflammation, but the evidence remains limited to pre-clinical work. Further research in broader animal models and human cohorts is needed before drawing clinical conclusions.

## Figures and Tables

**Figure 1 ijms-26-10269-f001:**
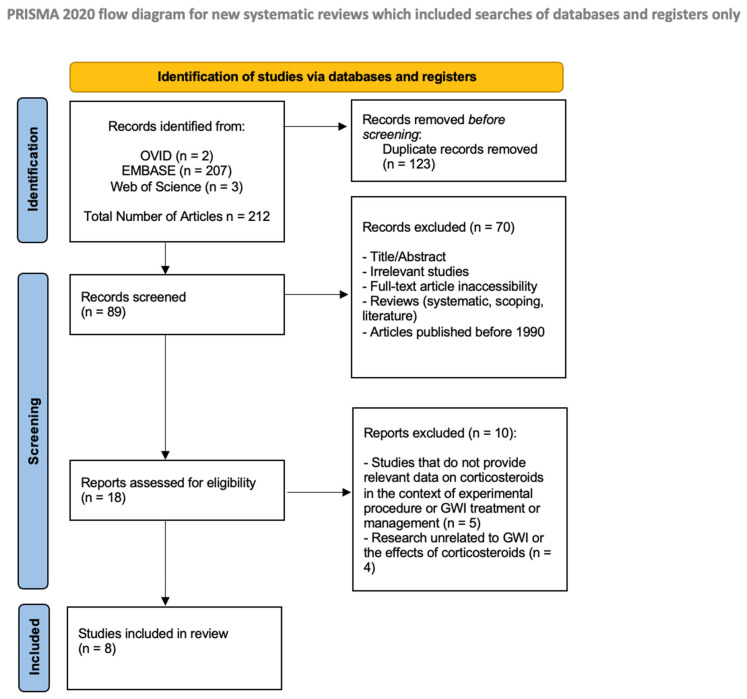
PRISMA diagram for selection criteria. PRISMA: Preferred Reporting Items for Systematic Reviews and Meta-Analyses; EMBASE: Excerpta Medica Database.

**Table 1 ijms-26-10269-t001:** Studies included in systematic review. GWI—Gulf War Illness, CORT—corticosterone, DFP—diisopropyl fluorophosphate, DEET—*N,N*-diethyl-meta-toluamide, PB—pyridostigmine bromide, LPS—lipopolysaccharide, CPO—chlorpyrifos oxon, AChE—acetylcholinesterase, CNS—central nervous system, PHY—physostidmine, MINO—minocycline.

Authors	Year	Central Question	Hypothesis	Groups	Treatment Intervention	Findings	Conclusions
O’Callaghan et al. [27]	2015	How does prior exposure to the stress hormone CORT enhance the CNS inflammatory response to subsequent exposure to DFP, a surrogate for the nerve agent sarin?	CORT exposure would sensitize the CNS, leading to an amplified neuroinflammatory response upon DFP exposure, thereby providing a potential animal model for GWI.	Adult male mice were divided into 5 mice per group.	**Group 1:** saline only **Group 2:** CORT in drinking water for one week prior to DFP exposure. **Group 3:** CORT + single injection of DFP **Group 4:** CORT + daily injections of PB and DEET **Group 5:** CORT + daily PB and DEET injections + single intraperitoneal injection of DFP. **Group 6:** CORT + administered MINO 30 min before DFP exposure + DFP	CORT pre-treatment enhanced DFP-induced neuroinflammation in multiple brain regions, with additional exacerbation observed when combined with PB and DEET exposure.	CORT primes the neuroinflammatory response to DFP, and this model, enhanced by exposure to PB and DEET, may provide insights into GWI pathophysiology.
Locker et al. [28]	2017	How does exposure to the stress hormone CORT enhance neuroinflammatory responses to organophosphates relevant to GWI, such as DFP and CPO, independently of AChE inhibition?	Prior exposure to CORT primes the brain’s immune system, leading to an exaggerated neuroinflammatory response upon subsequent exposure to these organophosphates, regardless of their AChE inhibitory properties.	Adult male mice were divided into 6 groups.	**Group 1:** saline only **Group 2:** CORT only **Group 3:** CORT + CPO **Group 4:** CORT + DFP **Group 5:** CORT + PB **Group 6:** CORT + PHY.	CORT pre-exposure amplified neuroinflammatory responses to organophosphates, increasing pro-inflammatory cytokines (IL-1β, TNFα), independent of acetylcholinesterase inhibition.	CORT primes the brain’s immune response to organophosphate exposure, enhancing neuroinflammation independently of AChE inhibition, possibly contributing to GWI.
Ashbrook et al. [29]	2018	How does prior stress exposure influence the neuroinflammatory and epigenetic response to a sarin surrogate in a mouse model of GWI?	Stress priming alters epigenetic regulation and enhances neuroinflammatory responses following exposure to DFP, a sarin surrogate, contributing to the persistent symptoms observed in GWI.	79 adult male mice divided into 4 experimental groups of 19–20 mice.	**Group 1:** saline in drinking water + saline injection. **Group 2:** CORT in drinking water + saline injection. **Group 3:** DFP in drinking water + DFP injection. **Group 4:** CORT in drinking water + DFP injection.	Stress priming with corticosterone enhanced neuroinflammatory responses, altered epigenetic regulation, and reduced myelinating oligodendrocytes following exposure to the DFP, suggesting a mechanism for persistent neurological dysfunction in GWI.	Stress + exposure to organophosphate compounds can lead to persistent epigenetic and neuroinflammatory changes, providing insights into the potential mechanisms underlying GWI.
Koo et al. [30]	2018	How does prior exposure to CORT, simulating physiological stress, influence neuroinflammatory responses and brain microstructure alterations following DFP exposure in a rat model of GWI?	Pre-exposure to CORT enhances DFP-induced neuroinflammation and leads to detectable changes in brain microstructure, as assessed by high-order diffusion MRI, thereby extending previous findings from mouse models to rats.	20 rats were divided into 4 experimental groups, with 5 rats per group.	**Group 1:** vehicle treatment (0.6% ethanol in drinking water) for 4 days, followed by a saline injection. **Group 2:** vehicle treatment for 4 days, followed by a single injection of DFP; 1.5 mg/kg, intraperitoneally. **Group 3:** CORT; 200 mg/L in 0.6% ethanol drinking water, for 4 days to mimic physiological stress, followed by a saline injection. **Group 4:** CORT treatment for 4 days, followed by a single DFP injection.	CORT pre-exposure amplified DFP-induced neuroinflammation, significantly increasing cytokine levels (TNFα, IL-6, IL-1β, etc.) in the cortex, while high-order diffusion MRI revealed distinct brain microstructural changes, particularly in the hippocampus and hypothalamus.	Stress exposure worsens DFP-induced neuroinflammation and brain changes, linking stress to GWI pathology.
Michalovicz et al. [30]	2019	How does CORT, combined with exposures to PB and DEET, affect peripheral cytokine expression in a mouse model of GWI?	Exposure to corticosterone, pyridostigmine, and DEET will lead to a reduction in peripheral cytokine levels.	Adult male mice were divided in 5 groups of 4–5 mice per group.	**Group 1:** saline injections only **Group 2:** CORT only in their drinking water to simulate stress. **Group 3:** CORT + DFP injection. **Group 4:** CORT + daily applications of PB and DEET, both Gulf War-related chemicals. **Group 5:** CORT + DFP injection, and daily applications of PB and DEET, representing a long-term GWI model.	CORT, combined with PB and DEET exposure, reduced peripheral cytokine expression, suggesting a dominant role for neuroinflammation in GWI.	Neuroinflammation plays a central role in GWI, and CORT, PB, and DEET exposure contribute to altering cytokine expression in this model.
Xu et al. [31]	2020	How does the combination of exposure to stressors, such as organophosphate agents and other GWI-related chemicals, impact gene expression profiles in specific tissues?	Chronic exposure to Gulf War-related agents, including PB, DEET, and/or nerve agents, would lead to significant alterations in gene expression, particularly in genes related to inflammation, immune response, and neuroinflammation.	Adult mice were split into 30 BXD RI strains with 2 mice per strain and sex and divided into 3 groups.	**Group 1:** saline in drinking water. **Group 2:** DFP injection. **Group 3:** CORT in water + DFP injection.	CORT pre-treatment enhanced the neuroinflammatory response to DFP exposure in mice.	CORT primes the neuroinflammatory response to DFP exposure in mice, supporting the development of an animal model for GWI and highlighting the potential role of stress hormones in exacerbating neuroinflammation.
Michalovicz et al. [32]	2021	Can propranolol, a β-adrenergic receptor blocker and anti-inflammatory medication, reduce brain cytokine expression in a long-term animal model of GWI?	Administering propranolol will decrease the expression of inflammatory cytokines in the brain of GWI model subjects, suggesting its potential as a therapeutic intervention for GWI-related neuroinflammation.	Adult male mice were divided into 5 groups with 5–7 mice per group.	**Group 1:** saline injections only **Group 2:** CORT only in their drinking water to simulate stress **Group 3:** CORT + DFP injection **Group 4:** CORT + LPS injection **Group 5:** CORT + DFP + LPS, representing a long-term GWI model. Additionally, a subset of these groups received propranolol treatment (20 mg/kg, intraperitoneally) either four or eleven days prior to the LPS challenge to assess its effects on neuroinflammation.	Propranolol reduced brain cytokine expression in the GWI model, specifically lowering levels of IL-1β, TNFα, and IL-6 in mice exposed to corticosterone, DFP, and LPS.	Propranolol mitigates neuroinflammation in a long-term GWI model, suggesting its potential as a therapeutic for GWI-related brain inflammation.
Cheng et al. [33]	2024	How do nerve agent exposure and physiological stress affect brain microstructure and immune responses following an inflammatory challenge in a long-term rat model of GWI?	Combined exposure to nerve agents and physiological stress leads to persistent alterations in brain microstructure and immune profiles, which become more pronounced after an inflammatory challenge, mirroring chronic symptoms observed in GWI.	86 adult male rats divided into 4 experimental groups with 17 rats each.	**Group 1:** saline in drinking water. **Group 2:** CORT was given in drinking water. **Group 3:** received CORT in drinking water followed by a DFP injection. **Group 4:** Same treatment as group 3 but was also given an LPS challenge.	Nerve agent and stress exposure in rats caused significant changes in brain microstructure and immune profiles, suggesting persistent neuroimmune activation linked to GWI.	Combined exposure to nerve agents and stress leads to long-term brain and immune system alterations, potentially contributing to the chronic symptoms of GWI.

## Data Availability

No new data were created or analyzed in this study. Data sharing is not applicable to this article.

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
