# Peer review of "Corticosteroid Usage in Modeling Gulf War Illness in Pre-Clinical Models: A Systematic Review"

_ijms, 2025, doi:10.3390/ijms262110269_

Round 1
Reviewer 1 Report
Comments and Suggestions for Authors
The review is fully structured and presents the current status of the effect of corticosteroids on the symptoms of Gulf War Illness, both directly and indirectly, integrating various approaches and molecular studies. However, for its acceptance, several suggestions must be addressed, as listed below:
L-15 Define CORT, since this is the first time it has appeared.
L-51 This reference indicates a greater number of studies; the text is superficial in supporting what is proposed.
L-61 While it clarifies the objective of the review, not why it is being conducted, and its relevance, it is recommended to structure the review's relevance at this point.
L-92 In the journal format, the figure lines should be below the figure.
L-106 Improve the quality of the table, as the letters are randomly cut off, such as Year.
L-126-128 References on this matter and references 23 are not sufficient to conclude this point. It is recommended to provide guidelines for other articles, even if they are not included in the inclusion criteria.
L-152 Improve the references on the relationship between JAK/STAT3 in the brain and its relationship with the pathology under study.
L-234 Improve the relationship between the Adamts9 gene and the symptoms of GWI, as this reference is superficial.
L-306 This section is superficial in justifying this unique therapeutic approach. There are other approaches in the literature, and a comparison is needed, even outside of a systematic review.
L-360 The conclusions are superficial; they do not integrate the symptoms with the molecular studies performed, and the relationship with JAK/STAT is not clear. Similarly, the results on Adamts9 are missing.
Reviewer 2 Report
Comments and Suggestions for Authors
This is a review article examining the involvement of corticosteroids in Gulf War Illness (GWI), for which no established treatment exists, using animal models. Relevant articles were extracted following PRISMA guidelines. As a result, nine papers describing model studies using mice or rats were selected. In each study, corticosteroids were used pre or during the establishment of the GWI models. That is, the extracted papers did not evaluate the effect of corticosteroids on GWI after onset. Therefore, this reviewer thinks that little insight can be gained regarding whether corticosteroids are useful for GWI patients. Sections 4.4 and 4.5 include evaluations of corticosteroid in the patients with GWI(in other words, after onset), which potentially mislead readers.
Overall, since the review content concerns the involvement of corticosteroids at the onset stage in animal models, the title should be changed accordingly.
Round 2
Reviewer 2 Report
Comments and Suggestions for Authors
The manuscript has been significantly improved.
Author Response
Thank you very much for your help! We believe that your review helped to significantly strengthen our manuscript.